# Dynamic Routing Between Capsules

**Sara Sabour**                    **Nicholas Frosst**

**Geoffrey E. Hinton**
Google Brain
Toronto
{sasabour, frosst, geoffhinton}@google.com

## Abstract

A capsule is a group of neurons whose activity vector represents the instantiation parameters of a specific type of entity such as an object or an object part. We use the length of the activity vector to represent the probability that the entity exists and its orientation to represent the instantiation parameters. Active capsules at one level make predictions, via transformation matrices, for the instantiation parameters of higher-level capsules. When multiple predictions agree, a higher level capsule becomes active. We show that a discrimininatively trained, multi-layer capsule system achieves state-of-the-art performance on MNIST and is considerably better than a convolutional net at recognizing highly overlapping digits. To achieve these results we use an iterative routing-by-agreement mechanism: A lower-level capsule prefers to send its output to higher level capsules whose activity vectors have a big scalar product with the prediction coming from the lower-level capsule.

## 1   Introduction

Human vision ignores irrelevant details by using a carefully determined sequence of fixation points to ensure that only a tiny fraction of the optic array is ever processed at the highest resolution. Introspection is a poor guide to understanding how much of our knowledge of a scene comes from the sequence of fixations and how much we glean from a single fixation, but in this paper we will assume that a single fixation gives us much more than just a single identified object and its properties. We assume that our multi-layer visual system creates a parse tree-like structure on each fixation, and we ignore the issue of how these single-fixation parse trees are coordinated over multiple fixations.

Parse trees are generally constructed on the fly by dynamically allocating memory. Following Hinton et al. [2000], however, we shall assume that, for a single fixation, a parse tree is carved out of a fixed multilayer neural network like a sculpture is carved from a rock. Each layer will be divided into many small groups of neurons called "capsules" (Hinton et al. [2011]) and each node in the parse tree will correspond to an active capsule. Using an iterative routing process, each active capsule will choose a capsule in the layer above to be its parent in the tree. For the higher levels of a visual system, this iterative process will be solving the problem of assigning parts to wholes.

The activities of the neurons within an active capsule represent the various properties of a particular entity that is present in the image. These properties can include many different types of instantiation parameter such as pose (position, size, orientation), deformation, velocity, albedo, hue, texture, etc. One very special property is the existence of the instantiated entity in the image. An obvious way to represent existence is by using a separate logistic unit whose output is the probability that the entity exists. In this paper we explore an interesting alternative which is to use the overall length of the vector of instantiation parameters to represent the existence of the entity and to force the orientation

of the vector to represent the properties of the entity[1]. We ensure that the length of the vector output of a capsule cannot exceed 1 by applying a non-linearity that leaves the orientation of the vector unchanged but scales down its magnitude.

The fact that the output of a capsule is a vector makes it possible to use a powerful dynamic routing mechanism to ensure that the output of the capsule gets sent to an appropriate parent in the layer above. Initially, the output is routed to all possible parents but is scaled down by coupling coefficients that sum to 1. For each possible parent, the capsule computes a "prediction vector" by multiplying its own output by a weight matrix. If this prediction vector has a large scalar product with the output of a possible parent, there is top-down feedback which increases the coupling coefficient for that parent and decreasing it for other parents. This increases the contribution that the capsule makes to that parent thus further increasing the scalar product of the capsule's prediction with the parent's output. This type of "routing-by-agreement" should be far more effective than the very primitive form of routing implemented by max-pooling, which allows neurons in one layer to ignore all but the most active feature detector in a local pool in the layer below. We demonstrate that our dynamic routing mechanism is an effective way to implement the "explaining away" that is needed for segmenting highly overlapping objects.

Convolutional neural networks (CNNs) use translated replicas of learned feature detectors. This allows them to translate knowledge about good weight values acquired at one position in an image to other positions. This has proven extremely helpful in image interpretation. Even though we are replacing the scalar-output feature detectors of CNNs with vector-output capsules and max-pooling with routing-by-agreement, we would still like to replicate learned knowledge across space. To achieve this, we make all but the last layer of capsules be convolutional. As with CNNs, we make higher-level capsules cover larger regions of the image. Unlike max-pooling however, we do not throw away information about the precise position of the entity within the region. For low level capsules, location information is "place-coded" by which capsule is active. As we ascend the hierarchy, more and more of the positional information is "rate-coded" in the real-valued components of the output vector of a capsule. This shift from place-coding to rate-coding combined with the fact that higher-level capsules represent more complex entities with more degrees of freedom suggests that the dimensionality of capsules should increase as we ascend the hierarchy.

## 2    How the vector inputs and outputs of a capsule are computed

There are many possible ways to implement the general idea of capsules. The aim of this paper is not to explore this whole space but simply to show that one fairly straightforward implementation works well and that dynamic routing helps.

We want the length of the output vector of a capsule to represent the probability that the entity represented by the capsule is present in the current input. We therefore use a non-linear **"squashing"** function to ensure that short vectors get shrunk to almost zero length and long vectors get shrunk to a length slightly below 1. We leave it to discriminative learning to make good use of this non-linearity.

$$\mathbf{v}_j = \frac{||\mathbf{s}_j||^2}{1 + ||\mathbf{s}_j||^2} \frac{\mathbf{s}_j}{||\mathbf{s}_j||} \tag{1}$$

where $\mathbf{v}_j$ is the vector output of capsule $j$ and $\mathbf{s}_j$ is its total input.

For all but the first layer of capsules, the total input to a capsule $\mathbf{s}_j$ is a weighted sum over all "prediction vectors" $\hat{\mathbf{u}}_{j|i}$ from the capsules in the layer below and is produced by multiplying the output $\mathbf{u}_i$ of a capsule in the layer below by a weight matrix $\mathbf{W}_{ij}$

$$\mathbf{s}_j = \sum_i c_{ij} \hat{\mathbf{u}}_{j|i} \,, \qquad \hat{\mathbf{u}}_{j|i} = \mathbf{W}_{ij} \mathbf{u}_i \tag{2}$$

where the $c_{ij}$ are coupling coefficients that are determined by the iterative dynamic routing process.

The coupling coefficients between capsule $i$ and all the capsules in the layer above sum to 1 and are determined by a "routing softmax" whose initial logits $b_{ij}$ are the log prior probabilities that capsule $i$

should be coupled to capsule $j$.

$$c_{ij} = \frac{\exp(b_{ij})}{\sum_k \exp(b_{ik})} \tag{3}$$

The log priors can be learned discriminatively at the same time as all the other weights. They depend on the location and type of the two capsules but not on the current input image[2]. The initial coupling coefficients are then iteratively refined by measuring the agreement between the current output $\mathbf{v}_j$ of each capsule, $j$, in the layer above and the prediction $\hat{\mathbf{u}}_{j|i}$ made by capsule $i$.

The agreement is simply the scalar product $a_{ij} = \mathbf{v}_j.\hat{\mathbf{u}}_{j|i}$. This agreement is treated as if it was a log likelihood and is added to the initial logit, $b_{ij}$ before computing the new values for all the coupling coefficients linking capsule $i$ to higher level capsules.

In convolutional capsule layers, each capsule outputs a local grid of vectors to each type of capsule in the layer above using different transformation matrices for each member of the grid as well as for each type of capsule.

---

**Procedure 1** Routing algorithm.

---

1: **procedure** ROUTING($\hat{\boldsymbol{u}}_{j|i}, r, l$)
2:     for all capsule $i$ in layer $l$ and capsule $j$ in layer $(l+1)$: $b_{ij} \leftarrow 0$.
3:     **for** $r$ iterations **do**
4:         for all capsule $i$ in layer $l$: $\mathbf{c}_i \leftarrow \texttt{softmax}(\mathbf{b}_i)$            ▷ `softmax` computes Eq. 3
5:         for all capsule $j$ in layer $(l+1)$: $\mathbf{s}_j \leftarrow \sum_i c_{ij}\hat{\mathbf{u}}_{j|i}$
6:         for all capsule $j$ in layer $(l+1)$: $\mathbf{v}_j \leftarrow \texttt{squash}(\mathbf{s}_j)$          ▷ `squash` computes Eq. 1
7:         for all capsule $i$ in layer $l$ and capsule $j$ in layer $(l+1)$: $b_{ij} \leftarrow b_{ij} + \hat{\mathbf{u}}_{j|i}.\mathbf{v}_j$
        **return** $\mathbf{v}_j$

---

## 3 Margin loss for digit existence

We are using the length of the instantiation vector to represent the probability that a capsule's entity exists. We would like the top-level capsule for digit class $k$ to have a long instantiation vector if and only if that digit is present in the image. To allow for multiple digits, we use a separate margin loss, $L_k$ for each digit capsule, $k$:

$$L_k = T_k \ \max(0, m^+ - ||\mathbf{v}_k||)^2 + \lambda \ (1 - T_k) \ \max(0, ||\mathbf{v}_k|| - m^-)^2 \tag{4}$$

where $T_k = 1$ iff a digit of class $k$ is present[3] and $m^+ = 0.9$ and $m^- = 0.1$. The $\lambda$ down-weighting of the loss for absent digit classes stops the initial learning from shrinking the lengths of the activity vectors of all the digit capsules. We use $\lambda = 0.5$. The total loss is simply the sum of the losses of all digit capsules.

## 4 CapsNet architecture

A simple CapsNet architecture is shown in Fig. 1. The architecture is shallow with only two convolutional layers and one fully connected layer. Conv1 has 256, $9 \times 9$ convolution kernels with a stride of 1 and ReLU activation. This layer converts pixel intensities to the activities of local feature detectors that are then used as inputs to the *primary* capsules.

The primary capsules are the lowest level of multi-dimensional entities and, from an inverse graphics perspective, activating the primary capsules corresponds to inverting the rendering process. This is a very different type of computation than piecing instantiated parts together to make familiar wholes, which is what capsules are designed to be good at.

The second layer (PrimaryCapsules) is a convolutional capsule layer with 32 channels of convolutional 8D capsules (*i.e.* each primary capsule contains 8 convolutional units with a $9 \times 9$ kernel and a stride of 2). Each primary capsule output sees the outputs of all $256 \times 81$ Conv1 units whose receptive

Figure 1: A simple CapsNet with 3 layers. This model gives comparable results to deep convolutional networks (such as Chang and Chen [2015]). The length of the activity vector of each capsule in DigitCaps layer indicates presence of an instance of each class and is used to calculate the classification loss. $\mathbf{W}_{ij}$ is a weight matrix between each $\mathbf{u}_i, i \in (1, 32 \times 6 \times 6)$ in PrimaryCapsules and $\mathbf{v}_j, j \in (1, 10)$.

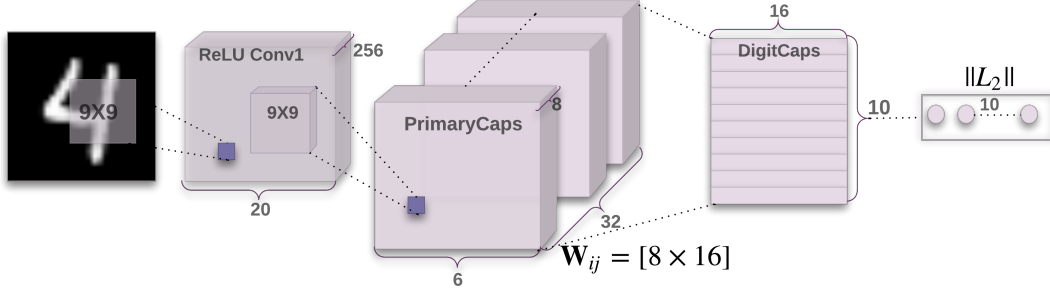

Figure 2: Decoder structure to reconstruct a digit from the DigitCaps layer representation. The euclidean distance between the image and the output of the Sigmoid layer is minimized during training. We use the true label as reconstruction target during training.

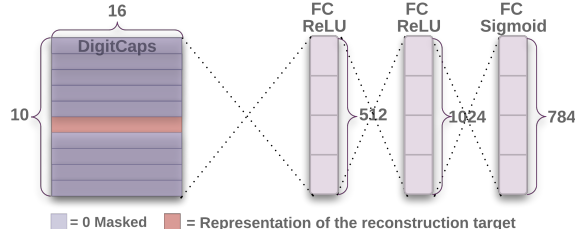

fields overlap with the location of the center of the capsule. In total PrimaryCapsules has $[32 \times 6 \times 6]$ capsule outputs (each output is an 8D vector) and each capsule in the $[6 \times 6]$ grid is sharing their weights with each other. One can see PrimaryCapsules as a Convolution layer with Eq. 1 as its block non-linearity. The final Layer (DigitCaps) has one 16D capsule per digit class and each of these capsules receives input from all the capsules in the layer below.

We have routing only between two consecutive capsule layers (e.g. PrimaryCapsules and DigitCaps). Since Conv1 output is 1D, there is no orientation in its space to agree on. Therefore, no routing is used between Conv1 and PrimaryCapsules. All the routing logits ($b_{ij}$) are initialized to zero. Therefore, initially a capsule output ($\mathbf{u}_i$) is sent to all parent capsules ($\mathbf{v}_0...\mathbf{v}_9$) with equal probability ($c_{ij}$).
Our implementation is in TensorFlow (Abadi et al. [2016]) and we use the Adam optimizer (Kingma and Ba [2014]) with its TensorFlow default parameters, including the exponentially decaying learning rate, to minimize the sum of the margin losses in Eq. 4.

### 4.1 Reconstruction as a regularization method

We use an additional reconstruction loss to encourage the digit capsules to encode the instantiation parameters of the input digit. During training, we mask out all but the activity vector of the correct digit capsule. Then we use this activity vector to reconstruct the input image. The output of the digit capsule is fed into a decoder consisting of 3 fully connected layers that model the pixel intensities as described in Fig. 2. We minimize the sum of squared differences between the outputs of the logistic units and the pixel intensities. We scale down this reconstruction loss by 0.0005 so that it does not dominate the margin loss during training. As illustrated in Fig. 3 the reconstructions from the 16D output of the CapsNet are robust while keeping only important details.

Figure 3: Sample MNIST test reconstructions of a CapsNet with 3 routing iterations. $(l, p, r)$ represents the label, the prediction and the reconstruction target respectively. The two rightmost columns show two reconstructions of a failure example and it explains how the model confuses a 5 and a 3 in this image. The other columns are from correct classifications and shows that model preserves many of the details while smoothing the noise.

| $(l,p,r)$ | $(2,2,2)$ | $(5,5,5)$ | $(8,8,8)$ | $(9,9,9)$ | $(5,3,5)$ | $(5,3,3)$ |
|---|---|---|---|---|---|---|
| Input | | | | | | |
| Output | | | | | | |

Table 1: CapsNet classification test accuracy. The MNIST average and standard deviation results are reported from 3 trials.

| Method | Routing | Reconstruction | MNIST (%) | MultiMNIST (%) |
|---|---|---|---|---|
| Baseline | - | - | 0.39 | 8.1 |
| CapsNet | 1 | no | $0.34_{\pm 0.032}$ | - |
| CapsNet | 1 | yes | $0.29_{\pm 0.011}$ | 7.5 |
| CapsNet | 3 | no | $0.35_{\pm 0.036}$ | - |
| CapsNet | 3 | yes | $\mathbf{0.25}_{\pm 0.005}$ | **5.2** |

# 5 Capsules on MNIST

Training is performed on $28 \times 28$ MNIST (LeCun et al. [1998]) images that have been shifted by up to 2 pixels in each direction with zero padding. No other data augmentation/deformation is used. The dataset has 60K and 10K images for training and testing respectively.

We test using a single model without any model averaging. Wan et al. [2013] achieves 0.21% test error with ensembling and augmenting the data with rotation and scaling. They achieve 0.39% without them. We get a low test error ($\mathbf{0.25}$%) on a 3 layer network previously only achieved by deeper networks. Tab. 1 reports the test error rate on MNIST for different CapsNet setups and shows the importance of routing and reconstruction regularizer. Adding the reconstruction regularizer boosts the routing performance by enforcing the pose encoding in the capsule vector.

The baseline is a standard CNN with three convolutional layers of $256, 256, 128$ channels. Each has 5x5 kernels and stride of 1. The last convolutional layers are followed by two fully connected layers of size $328, 192$. The last fully connected layer is connected with dropout to a 10 class softmax layer with cross entropy loss. The baseline is also trained on 2-pixel shifted MNIST with Adam optimizer. The baseline is designed to achieve the best performance on MNIST while keeping the computation cost as close as to CapsNet. In terms of number of parameters the baseline has 35.4M while CapsNet has 8.2M parameters and 6.8M parameters without the reconstruction subnetwork.

## 5.1 What the individual dimensions of a capsule represent

Since we are passing the encoding of only one digit and zeroing out other digits, the dimensions of a digit capsule should learn to span the space of variations in the way digits of that class are instantiated. These variations include stroke thickness, skew and width. They also include digit-specific variations such as the length of the tail of a 2. We can see what the individual dimensions represent by making use of the decoder network. After computing the activity vector for the correct digit capsule, we can feed a perturbed version of this activity vector to the decoder network and see how the perturbation affects the reconstruction. Examples of these perturbations are shown in Fig. 4. We found that one dimension (out of 16) of the capsule almost always represents the width of the digit. While some dimensions represent combinations of global variations, there are other dimensions that represent

Figure 4: Dimension perturbations. Each row shows the reconstruction when one of the 16 dimensions in the DigitCaps representation is tweaked by intervals of 0.05 in the range $[-0.25, 0.25]$.

| | |
|---|---|
| Scale and thickness | 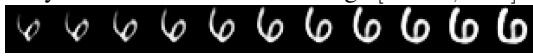 |
| Localized part |  |
| Stroke thickness |  |
| Localized skew |  |
| Width and translation |  |
| Localized part |  |

variation in a localized part of the digit. For example, different dimensions are used for the length of the ascender of a 6 and the size of the loop.

## 5.2 Robustness to Affine Transformations

Experiments show that each DigitCaps capsule learns a more robust representation for each class than a traditional convolutional network. Because there is natural variance in skew, rotation, style, etc in hand written digits, the trained CapsNet is moderately robust to small affine transformations of the training data.

To test the robustness of CapsNet to affine transformations, we trained a CapsNet and a traditional convolutional network (with MaxPooling and DropOut) on a padded and translated MNIST training set, in which each example is an MNIST digit placed randomly on a black background of $40 \times 40$ pixels. We then tested this network on the affNIST[4] data set, in which each example is an MNIST digit with a random small affine transformation. Our models were never trained with affine transformations other than translation and any natural transformation seen in the standard MNIST. An under-trained CapsNet with early stopping which achieved 99.23% accuracy on the expanded MNIST test set achieved 79% accuracy on the affnist test set. A traditional convolutional model with a similar number of parameters which achieved similar accuracy (99.22%) on the expanded mnist test set only achieved 66% on the affnist test set.

## 6 Segmenting highly overlapping digits

Dynamic routing can be viewed as a parallel attention mechanism that allows each capsule at one level to attend to some active capsules at the level below and to ignore others. This should allow the model to recognize multiple objects in the image even if objects overlap. Hinton et al. propose the task of segmenting and recognizing highly overlapping digits (Hinton et al. [2000] and others have tested their networks in a similar domain (Goodfellow et al. [2013], Ba et al. [2014], Greff et al. [2016]). The routing-by-agreement should make it possible to use a prior about the shape of objects to help segmentation and it should obviate the need to make higher-level segmentation decisions in the domain of pixels.

## 6.1 MultiMNIST dataset

We generate the MultiMNIST training and test dataset by overlaying a digit on top of another digit from the same set (training or test) but different class. Each digit is shifted up to 4 pixels in each direction resulting in a $36 \times 36$ image. Considering a digit in a $28 \times 28$ image is bounded in a $20 \times 20$ box, two digits bounding boxes on average have 80% overlap. For each digit in the MNIST dataset we generate 1K MultiMNIST examples. So the training set size is 60M and the test set size is 10M.

Figure 5: Sample reconstructions of a CapsNet with 3 routing iterations on MultiMNIST test dataset. The two reconstructed digits are overlayed in green and red as the lower image. The upper image shows the input image. L:$(l_1, l_2)$ represents the label for the two digits in the image and R:$(r_1, r_2)$ represents the two digits used for reconstruction. The two right most columns show two examples with wrong classification reconstructed from the label and from the prediction (P). In the $(2, 8)$ example the model confuses $8$ with a $7$ and in $(4, 9)$ it confuses $9$ with $0$. The other columns have correct classifications and show that the model accounts for all the pixels while being able to assign one pixel to two digits in extremely difficult scenarios (column $1 - 4$). Note that in dataset generation the pixel values are clipped at $1$. The two columns with the (*) mark show reconstructions from a digit that is neither the label nor the prediction. These columns suggests that the model is not just finding the best fit for all the digits in the image including the ones that do not exist. Therefore in case of $(5, 0)$ it cannot reconstruct a $7$ because it knows that there is a $5$ and $0$ that fit best and account for all the pixels. Also, in case of $(8, 1)$ the loop of $8$ has not triggered $0$ because it is already accounted for by $8$. Therefore it will not assign one pixel to two digits if one of them does not have any other support.

| R:(2,7) L:(2,7) | R:(6,0) L:(6,0) | R:(6,8) L:(6,8) | R:(7,1) L:(7,1) | *R:(5,7) L:(5,0) | *R:(2,3) L:(4,3) | R:(2,8) L:(2,8) | R:P:(2,7) L:(2,8) |
|---|---|---|---|---|---|---|---|
| 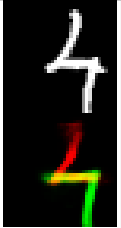  |   |   |   |   |   |   |   |

| R:(8,7) L:(8,7) | R:(9,4) L:(9,4) | R:(9,5) L:(9,5) | R:(8,4) L:(8,4) | *R:(0,8) L:(1,8) | *R:(1,6) L:(7,6) | R:(4,9) L:(4,9) | R:P:(4,0) L:(4,9) |
|---|---|---|---|---|---|---|---|
|   |   |   |   |   |   |   |   |

## 6.2 MultiMNIST results

Our 3 layer CapsNet model trained from scratch on MultiMNIST training data achieves higher test classification accuracy than our baseline convolutional model. We are achieving the same classification error rate of $5.0\%$ on highly overlapping digit pairs as the sequential attention model of Ba et al. [2014] achieves on a much easier task that has far less overlap ($80\%$ overlap of the boxes around the two digits in our case vs $< 4\%$ for Ba et al. [2014]). On test images, which are composed of pairs of images from the test set, we treat the two most active digit capsules as the classification produced by the capsules network. During reconstruction we pick one digit at a time and use the activity vector of the chosen digit capsule to reconstruct the image of the chosen digit (we know this image because we used it to generate the composite image). The only difference with our MNIST model is that we increased the period of the decay step for the learning rate to be $10\times$ larger because the training dataset is larger.

The reconstructions illustrated in Fig. 5 show that CapsNet is able to segment the image into the two original digits. Since this segmentation is not at pixel level we observe that the model is able to deal correctly with the overlaps (a pixel is on in both digits) while accounting for all the pixels. The position and the style of each digit is encoded in DigitCaps. The decoder has learned to reconstruct a digit given the encoding. The fact that it is able to reconstruct digits regardless of the overlap shows that each digit capsule can pick up the style and position from the votes it is receiving from PrimaryCapsules layer.

Tab. 1 emphasizes the importance of capsules with routing on this task. As a baseline for the classification of CapsNet accuracy we trained a convolution network with two convolution layers and two fully connected layers on top of them. The first layer has $512$ convolution kernels of size $9 \times 9$ and stride $1$. The second layer has $256$ kernels of size $5 \times 5$ and stride $1$. After each convolution layer the model has a pooling layer of size $2 \times 2$ and stride $2$. The third layer is a $1024$D fully connected layer. All three layers have ReLU non-linearities. The final layer of $10$ units is fully connected. We use the TensorFlow default Adam optimizer (Kingma and Ba [2014]) to train a sigmoid cross entropy loss on the output of final layer. This model has $24.56$M parameters which is $2$ times more parameters than CapsNet with $11.36$M parameters. We started with a smaller CNN ($32$ and $64$ convolutional kernels of $5 \times 5$ and stride of $1$ and a $512$D fully connected layer) and incrementally increased the width of the network until we reached the best test accuracy on a $10$K subset of the MultiMNIST data. We also searched for the right decay step on the $10$K validation set.

We decode the two most active DigitCaps capsules one at a time and get two images. Then by assigning any pixel with non-zero intensity to each digit we get the segmentation results for each digit.

# 7 Other datasets

We tested our capsule model on CIFAR10 and achieved $10.6\%$ error with an ensemble of $7$ models each of which is trained with $3$ routing iterations on $24 \times 24$ patches of the image. Each model has the same architecture as the simple model we used for MNIST except that there are three color channels and we used $64$ different types of primary capsule. We also found that it helped to introduce a "none-of-the-above" category for the routing softmaxes, since we do not expect the final layer of ten capsules to explain everything in the image. $10.6\%$ test error is about what standard convolutional nets achieved when they were first applied to CIFAR10 (Zeiler and Fergus [2013]).

One drawback of Capsules which it shares with generative models is that it likes to account for everything in the image so it does better when it can model the clutter than when it just uses an additional "orphan" category in the dynamic routing. In CIFAR-10, the backgrounds are much too varied to model in a reasonable sized net which helps to account for the poorer performance.

We also tested the exact same architecture as we used for MNIST on smallNORB (LeCun et al. [2004]) and achieved $2.7\%$ test error rate, which is on-par with the state-of-the-art (Cireşan et al. [2011]). The smallNORB dataset consists of 96x96 stereo grey-scale images. We resized the images to 48x48 and during training processed random 32x32 crops of them. We passed the central 32x32 patch during test.

We also trained a smaller network on the small training set of SVHN (Netzer et al. [2011]) with only 73257 images. We reduced the number of first convolutional layer channels to 64, the primary capsule layer to $16$ $6$D-capsules with $8$D final capsule layer at the end and achieved $4.3\%$ on the test set.

# 8 Discussion and previous work

For thirty years, the state-of-the-art in speech recognition used hidden Markov models with Gaussian mixtures as output distributions. These models were easy to learn on small computers, but they had a representational limitation that was ultimately fatal: The one-of-n representations they use are exponentially inefficient compared with, say, a recurrent neural network that uses distributed representations. To double the amount of information that an HMM can remember about the string it has generated so far, we need to square the number of hidden nodes. For a recurrent net we only need to double the number of hidden neurons.

Now that convolutional neural networks have become the dominant approach to object recognition, it makes sense to ask whether there are any exponential inefficiencies that may lead to their demise. A good candidate is the difficulty that convolutional nets have in generalizing to novel viewpoints. The ability to deal with translation is built in, but for the other dimensions of an affine transformation we have to chose between replicating feature detectors on a grid that grows exponentially with the number of dimensions, or increasing the size of the labelled training set in a similarly exponential way. Capsules (Hinton et al. [2011]) avoid these exponential inefficiencies by converting pixel intensities

into vectors of instantiation parameters of recognized fragments and then applying transformation matrices to the fragments to predict the instantiation parameters of larger fragments. Transformation matrices that learn to encode the intrinsic spatial relationship between a part and a whole constitute viewpoint invariant knowledge that automatically generalizes to novel viewpoints. Hinton et al. [2011] proposed transforming autoencoders to generate the instantiation parameters of the PrimaryCapsule layer and their system required transformation matrices to be supplied externally. We propose a complete system that also answers "how larger and more complex visual entities can be recognized by using agreements of the poses predicted by active, lower-level capsules".

Capsules make a very strong representational assumption: At each location in the image, there is at most one instance of the type of entity that a capsule represents. This assumption, which was motivated by the perceptual phenomenon called "crowding" (Pelli et al. [2004]), eliminates the binding problem (Hinton [1981a]) and allows a capsule to use a distributed representation (its activity vector) to encode the instantiation parameters of *the* entity of that type at a given location. This distributed representation is exponentially more efficient than encoding the instantiation parameters by activating a point on a high-dimensional grid and with the right distributed representation, capsules can then take full advantage of the fact that spatial relationships can be modelled by matrix multiplies.

Capsules use neural activities that vary as viewpoint varies rather than trying to eliminate viewpoint variation from the activities. This gives them an advantage over "normalization" methods like spatial transformer networks (Jaderberg et al. [2015]): They can deal with multiple different affine transformations of different objects or object parts at the same time.

Capsules are also very good for dealing with segmentation, which is another of the toughest problems in vision, because the vector of instantiation parameters allows them to use routing-by-agreement, as we have demonstrated in this paper. The importance of dynamic routing procedure is also backed by biologically plausible models of invarient pattern recognition in the visual cortex. Hinton [1981b] proposes dynamic connections and canonical object based frames of reference to generate shape descriptions that can be used for object recognition. Olshausen et al. [1993] improves upon Hinton [1981b] dynamic connections and presents a biologically plausible, position and scale invariant model of object representations.

Research on capsules is now at a similar stage to research on recurrent neural networks for speech recognition at the beginning of this century. There are fundamental representational reasons for believing that it is a better approach but it probably requires a lot more small insights before it can out-perform a highly developed technology. The fact that a simple capsules system already gives unparalleled performance at segmenting overlapping digits is an early indication that capsules are a direction worth exploring.

**Acknowledgement.** Of the many who provided us with constructive comments, we are specially grateful to Robert Gens, Eric Langlois, Vincent Vanhoucke, Chris Williams, and the reviewers for their fruitful comments and corrections.

## Footnotes

[1]This makes biological sense as it does not use large activities to get accurate representations of things that probably don't exist.

[2]For MNIST we found that it was sufficient to set all of these priors to be equal.

[3]We do not allow an image to contain two instances of the same digit class. We address this weakness of capsules in the discussion section.

[4]Available at http://www.cs.toronto.edu/~tijmen/affNIST/.

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

## A    How many routing iterations to use?

In order to experimentally verify the convergence of the routing algorithm we plot the average change in the routing logits at each routing iteration. Fig. A.1 shows the average $b_{ij}$ change after each routing iteration. Experimentally we observe that there is negligible change in the routing by $5$ iteration from the start of training. Average change in the $2^{nd}$ pass of the routing settles down after 500 epochs of training to 0.007 while at routing iteration 5 the logits only change by $1e-5$ on average.

Figure A.1: Average change of each routing logit ($b_{ij}$) by each routing iteration. After 500 epochs of training on MNIST the average change is stabilized and as it shown in right figure it decreases almost linearly in log scale with more routing iterations.

<div align="center">(a) During training.&emsp;&emsp;&emsp;&emsp;&emsp;&emsp;(b) Log scale of final differences.</div>

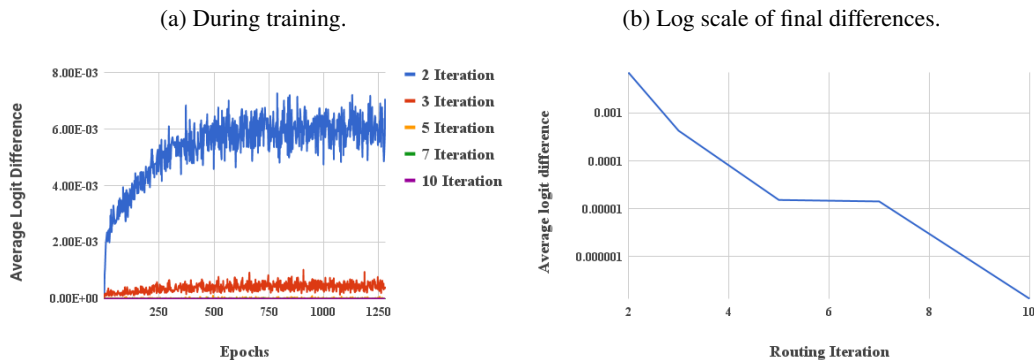

We observed that in general more routing iterations increases the network capacity and tends to overfit to the training dataset. Fig. A.2 shows a comparison of Capsule training loss on Cifar10 when trained with 1 iteration of routing vs 3 iteration of routing. Motivated by Fig. A.2 and Fig. A.1 we suggest 3 iteration of routing for all experiments.

Figure A.2: Traning loss of CapsuleNet on cifar10 dataset. The batch size at each training step is 128. The CapsuleNet with 3 iteration of routing optimizes the loss faster and converges to a lower loss at the end.

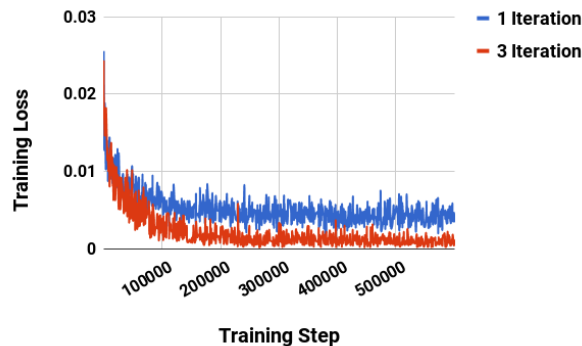

