[Reviews · NeurIPS 2017]

Reviewer 1



Overview :this paper introduces a dynamic routing process for connecting layers in a feedforward neural net, as described in Procedure 1 on p 3. The key idea here is that the coupling coeff c_ij between unit i and unit j is computed dynamically (layerwise), taking into account the agreement between the output v_j of unit j, and the prediction from unit i \hat{u}_{{j|i}. This process is iterates between each layer l and l+1, but does not (as far as I can tell) spread further back. Another innovation used in the paper is a form of nonlinearity as in eq 1 for units which uses the length of the capsule output v_j to encode strength of activity, and the direction of v_j to encode the values of the capsule parameters. A shallow CapsNet model is trained on MNIST, and obtains very good performance (a check of the MNIST leaderboard shows best performance of 0.23 obtained with a committee of deep conv nets), cf performance in Table 1. I regard this paper as very interesting, as it has successfully married the capsules idea with conv nets, and makes use of the dynamic routing capabilities. The results on highly overlapping digits (sec 6) are also impressive. This idea could rally take off and be heavily used. One major question about the paper is the convergence of the iteration in Proc 1. It would be highly desirable to prove that this converges (if it does so). This issue is as far as I can tell not discussed in the paper. It would also be very nice to see an example of the dynamic routing process in action, especially to see how a unit whose \hat{u}_{j|i} is out of agreement with the others gets suppressed. This process of assessing agreement of bottom-up contributions is reminiscent of the work in Rao and Ballard (Predictive coding in the visual cortex: a functional interpretation of some extra-classical receptive-field effects, Nature Neurosci. 2(1) 79-87, 1999), esp the example where local bar detectors all predict the same longer bar. One could e.g. look at how activations of 3 units in a line --- compare to one that is disoriented, e.g. |-- in your network. I would encourage the authors to publish their code (this can be following acceptance). This would greatly facilitate uptake by the community. Quality: As far as I can tell this is technically sound. Clarity: The writing is clear but my detailed comments give a number of suggestions to make the presentation less idiosyncratic. Originality: The idea of capsules is not new, but this paper seems to have achieved a step change in their performance and moved beyond the "transforming autoencoder" framework in which they were trained. Significance: The level of performance that can be achieved with a relatively shallow network is impressive. Given the interest in object recognition etc it is possible that this dynamic routing idea could really take off. Other points: p 1 Need to ref Hinton et al (20111) when you first mention "capsules" in the text. p2 and Procedure 1:I strongly recommend "squashing" to "squidging" as it is a term that has been used in the NN literature (and sounds more mathematical). p 2 The constraint that the sum_j c_{ij} = 1 is reminiscent of the "single parent constraint" in Hinton et al (2000). It would be good to discuss this more. p 3. The 0.5 in eq (4) seems like a hack, although it would look better if you introduced a weighting parameter alpha and set it to 0.5. Also I don't understand why you are not just using softmax loss (for a single digit classifier) on the ||v_j||'s (as they lie in 0,1). (For the multiclass case you can use multiple independent sigmoids.) p 3 I recommend "PrimaryCapsules" rather than "PrimaryCaps" as Caps sounds like CAPS. p 5 Lee et al [2016] baseline -- justify why this is an appropriate baseline. p 5 sec 5.1 Tab. 4 -> Fig. 4 p 6 Figure 4 caption: "Each column" -> "Each row". p 6 Sec 6 first para -- it is confusing to talk about your performance level before you have introduced the task -- leave this for later. p 6 sec 6: note a heavily overlapping digits task was used in Hinton et al (2000). p 8. It would be better to put the related work earlier, e.g. after sec 4. p 9. The ordering of the references (as they appear in the paper) is unusual and makes them hard to find. Suggest alphabetical or numbered in order of use. These are standard bibtex options.

Reviewer 2



This paper presents CapsNet, a multi-layer capsule system (similar to a deep network with multiple layers) and demonstrate the effectiveness of their approach on the MNIST dataset and demonstrate ability to handle multiple overlapping digits as well as affine perturbations. A capsule, as proposed by "Transforming Auto-encoders" 2011 by Hinton, is a set of neurons whose activity (represented as a vector) suggests the presence of a visual object (measured by the magnitude of activity vector) and the object parameters (represented by the orientation of the vector). A multi-layer capsule system consists of several layers, where each layer is a set of capsules, and hence architecturally it bears significantly resemblance to a deep network with several convolution and pooling layers. A key difference is that instead of a scalar output of a convolution, capsules have a vector output, and instead of max-pooling to route information from low level layers to high level layers, it uses dynamic routing based on the activation of the capsules (magnitude of the activity vector). For training, they use a 'margin loss' over all digits classes as well as a reconstruction loss (as a regularity to reconstruct the image) from the last activity layer. This paper uses a 3 layered capsule system trained on MNIST datasets with translation perturbations, and significantly outperformed a similarly sized CNN with max-pooling and dropout on affNIST data (MNIST with affine perturbations). A multiMNIST dataset is presented with images of overlapping digits with the task of identifying both the digits; the proposed capsule system successfully segmented the images into the images of both digits. Evaluation on CIFAR10 dataset achieved performance similar to early CNN evaluations (15.7% error). Strengths * Paper presents significant refinements to the 2011's capsule system (Transforming Auto-encoders), achieving strong performance with shallow networks. * The trained network demonstrated greater robustness to affine perturbations compared to CNNs with max-pooling and dropout, suggesting possibility of better generalization. This was attributed to the dynamic routing of information in capsule system; this line of research would indeed be useful to the NIPS community. Weaknesses * (Primary concern) Paper is too dense and is not very easy to follow; multiple reads were required to grasp the concepts and contribution. I would strongly recommend simplifying the description and explaining the architecture and computations better; Figure 7, Section 8 as well as lines 39-64 can be reduced to gain more space. * While the MNIST and CIFAR experiments is promising but they are not close to the state-of-art methods. It is not obvious if such explicitly dynamic routing is required to address the problem OR if recent advances such as residual units that have enabled significantly deeper networks can implicitly capture routing even with simple schemes such as a max-pooling. It would be good if authors can share their insights on this.

Reviewer 3



Quality The presented work is technically sound, though ideas as such are not novel. However, the particular implementation presented is. The authors discuss strengths and weaknesses of their presented approach, showing promising results on MNIST variants and drawbacks on more realistic tasks like Cifar10. Clarity The paper is well-organized, but some details are confusing or unclear. - Discuss difference to original capsule work. - Why are 1-3 refinement iterations chosen, what happens after more iterations? - How many iterations were necessary for Cifar10? - Compare the computational cost of baseline and capsules, as well as the cost of the refinement steps. - What happens when the invariant sum over the coupled prediction vectors in equation (2) and the associated non-linearity are replaced by a simple linear layer and standard non-linearity? - line 135: "We test using a single model with no ... data augmentation". A couple of lines before, the authors mention they do moderately augment data with shifts. Why do shifts improve performance, given that the authors claim capsules are designed to be robust to such variations? Originality The presented work is original as it introduces a new routing principle for capsule networks. However, novelty with respect to classical capsules should be discussed more clearly. Relevant related work, either dealing with separating filter response into magnitude and orientation, estimating keypoints or doing locally adaptive filtering, as opposed to global normalization of STNs: https://arxiv.org/abs/1701.01833 https://arxiv.org/abs/1703.06211 https://arxiv.org/abs/1612.04642 https://arxiv.org/abs/1706.00598 https://arxiv.org/abs/1605.01224 https://arxiv.org/abs/1605.09673 Significance The presented results are not very strong and it is hard to say how significant the findings are, as the authors do not thoroughly investigate more interesting domains than digits. Performance on MNIST is a very limited metric, given that: i) Saturated at this point ii) Scattering has shown that local invariance wrt deformation, translation and rotation is enough to achieve very good performance iii) It lacks ambiguity and many other properties that make natural images so challenging, especially the assumption of only one entity per location becomes questionable under clutter The robustness results on affine and overlapping MNIST are promising, but should be validated on more realistic tasks with more challenging local statistics. It would be great if the authors would provide the reader with insight into strengths and weaknesses on more realistic problems. Some suggestions: i) More thorough evaluation + visualisations on Cifar10. The results seem weak for now, but might shed some light on failure modes and work to be accomplished by follow-up papers ii) Check if affine robustness holds for Cifar10 as well to similar degree, this would change my vote on the paper iii) The iterative Tagger (Graeff et al.) might give some inspiration for additional experiments with more natural ambiguity and should be discussed in related work as well A strong analysis on the drawbacks of the presented method and open problems would make this a very useful paper, but in its current form, it is hard to tell what the suggested method achieves beyond being potentially more representationally efficient and robust on variants of MNIST.